# Environmental sampling for typhoidal *Salmonellas* in household and surface waters in Nepal identifies potential transmission pathways

**Christopher LeBoa**[1,2☯*], **Sneha Shrestha**[3☯], **Jivan Shakya**[4☯], **Shiva Ram Naga**[3,5], **Sony Shrestha**[3], **Mudita Shakya**[3], **Alexander T. Yu**[1], **Rajeev Shrestha**[3,5], **Krista Vaidya**[3], **Nishan Katuwal**[3,5], **Kristen Aiemjoy**[6,7], **Isaac I. Bogoch**[8], **Christopher B. Uzzell**[9], **Denise O. Garrett**[10], **Stephen P. Luby**[1], **Jason R. Andrews**[1☯], **Dipesh Tamrakar**[3,5☯]

1 Stanford University, Division of Infectious Diseases and Geographic Medicine, Stanford, California, United States of America, 2 University of California Berkeley, Department of Environmental Health Sciences, Berkeley, California, United States of America, 3 Research and Development Division, Dhulikhel Hospital Kathmandu University Hospital, Kavre, Nepal, 4 Institute for Research in Science and Technology, Lalitpur, Nepal, 5 Center for Infectious Disease Research and Surveillance, Dhulikhel Hospital Kathmandu University Hospital, Kavre, Nepal, 6 University of California Davis, Division of Public Health Sciences, California, United States of America, 7 Mahidol University Faculty of Tropical Medicine, Department of Microbiology and Immunology, Bangkok, Thailand, 8 Toronto General Hospital, Division of Infectious Diseases, Toronto, Canada, and Department of Medicine, University of Toronto, Toronto Canada, 9 Imperial College London, School of Public Health, Norfolk Place, London, United Kingdom, 10 Sabin Vaccine Institute, Applied Epidemiology Section, Washington, DC, United States of America

☯ These authors contributed equally to this work.
* cleboa@berkeley.edu

**Data Availability Statement:** The data used for this publication has been uploaded to Dryad (DOI: 10.6078/D1ZD84)

## Abstract

### Introduction

*Salmonella* Typhi and *Salmonella* Paratyphi, fecal-oral transmitted bacterium, have temporally and geographically heterogeneous pathways of transmission. Previous work in Kathmandu, Nepal implicated stone waterspouts as a dominant transmission pathway after 77% of samples tested positive for *Salmonella* Typhi and 70% for *Salmonella* Paratyphi. Due to a falling water table, these spouts no longer provide drinking water, but typhoid fever persists, and the question of the disease's dominant pathway of transmission remains unanswered.

### Methods

We used environmental surveillance to detect *Salmonella* Typhi and *Salmonella* Paratyphi A DNA from potential sources of transmission. We collected 370, 1L drinking water samples from a population-based random sample of households in the Kathmandu and Kavre Districts of Nepal between February and October 2019. Between November 2019 and July 2021, we collected 380, 50mL river water samples from 19 sentinel sites on a monthly interval along the rivers leading through the Kathmandu and Kavre Districts. We processed drinking water samples using a single qPCR and processed river water samples using differential centrifugation and qPCR at 0 and after 16 hours of liquid culture enrichment. A 3-cycle

(URL: https://datadryad.org/stash/share/PCCHoAMxJjFX709DH6SGTkkTpPbi1QL5DZksRsYzFsE). The protocols used for this publication are available on protocols.io and linked in the text and here(DOI: dx.doi.org/10.17504/protocols.io.ewov1o6bplr2/v1). The code used for cleaning data, analysis and visualizations is availible on github and linked in the text.

**Funding:** This work was supported by Bill & Melinda Gates Foundation, Seattle, WA [grant number INV-000572] received by JA, KA, AY and SL (url: https://www.gatesfoundation.org/); and Stanford University Center for Innovation in Global Health Seed Grant received by JA, AY and KA. The funders had no role in study design, data collection and analysis, decision to publish, or preparation of the manuscript. No authors receive their salary from the funding organizations.

**Competing interests:** The authors declare they have no actual or potential competing financial interests. IIB consults to BlueDot, a social benefit corporation that tracks emerging infectious diseases, and to the NHL Players' Association.

threshold (Ct) decrease of *Salmonella* Typhi or *Salmonella* Paratyphi, pre- and post-enrichment, was used as evidence of growth. We also performed structured observations of human-environment interactions to understand pathways of potential exposure.

## Results

Among 370 drinking water samples, *Salmonella* Typhi was detected in 7 samples (1.8%) and *Salmonella* Paratyphi A was detected in 4 (1.0%) samples. Among 380 river water samples, *Salmonella* Typhi was detected in 171 (45%) and *Salmonella* Paratyphi A was detected in 152 (42%) samples. Samples located upstream of the Kathmandu city center were positive for *Salmonella* Typhi 12% of the time while samples from locations in and downstream were positive 58% and 67% of the time respectively. Individuals were observed bathing, washing clothes, and washing vegetables in the rivers.

## Implications

These results suggest that drinking water was not the dominant pathway of transmission of *Salmonella* Typhi and *Salmonella* Paratyphi A in the Kathmandu Valley in 2019. The high degree of river water contamination and its use for washing vegetables raises the possibility that river systems represent an important source of typhoid exposure in Kathmandu.

### Author summary

Understanding the dominant route of transmission of a pathogen is important for designing and implementing effective control strategies. *Salmonella* Typhi and Paratyphi A, which cause enteric fever, infect approximately 10 million people and cause over 100,000 deaths annually. In the Kathmandu Valley, prior work suggested ancient stone spouts used for drinking water were often contaminated and driving transmission of the diseases. However, many of these spouts no longer function, and people are still getting sick, suggesting other possible dominant pathways for enteric fever transmission. We tested drinking water from households in this area as well as local river water and found that only 7 drinking water samples were positive for *Salmonella* Typhi and 4 were positive for *Salmonella* Paratyphi A. We also tested river water and found many samples (>40%) tested positive for these bacteria. River water samples were not often positive upstream of Kathmandu city center (12% positive for *Salmonella* Typhi) but were often positive within the city center (58% positive for *Salmonella* Typhi) and in rural areas up to 10 km downstream of the city (67% positive for *Salmonella* Typhi). During sample collection, individuals were observed interacting with rivers by walking in them, washing clothes and washing vegetables for sale in markets. This study shows that drinking water may not be a primary driver of enteric fever transmission in the Kathmandu Valley, but that sewage contaminated river water may be a way disease transmits into the wider population.

## Introduction

Enteric fever, a fecal-oral transmitted febrile illness caused by *Salmonella enterica* serovars Typhi and Paratyphi A, causes 10 to 20 million illnesses and over 100,00 deaths annually [1]. Improvements in the sanitation infrastructure in much of Europe and North America in the

20th century eliminated enteric fever transmission in those areas [2]. However, enteric fever remains highly endemic in many areas today, especially in Africa and South Asia [1]. In Nepal, a recent population-based study reported a high incidence of typhoid fever in Kathmandu (330 cases per 100,000 person-years) and Kavre (268 cases per 100,000 person-years) with a sero-incidence of 6.6 per 100 person years [3,4]. Increasing drug resistance among *Salmonella* Typhi and *Salmonella* Paratyphi strains are rendering antibiotic treatment for enteric fever less effective [5,6]. The World Health Organization (WHO) recently prequalified two typhoid conjugate vaccines (TCV), one of which was shown to be effective in reducing typhoid incidence in three randomized trials [7–9]. In a Phase 3 trial of TCV conducted in Nepal, vaccine recipients had an typhoid fever incidence of 79 cases per 100,000 person-years, indicating that even effective vaccines alone cannot end transmission [7]. Ending transmission must also address environmental and political issues. Clarifying pathways of *Salmonella* Typhi and *Salmonella* Paratyphi transmission permits focused interventions to interrupt transmission. Environmental sampling has emerged as a pragmatic, low-cost, non-invasive approach to detecting different pathogens, including *S.* Typhi and *S.* Paratyphi, across different mediums to help identify modalities by which people are exposed [10,11].

*Salmonella* Typhi and Paratyphi A are known to have spatial and temporal variation in their dominant route of transmission [12]. Within Kathmandu and Kavre Districts of Nepal, enteric fever incidence rates vary widely by ward, a sub-administrative unit, with S. Typhi and S. Paratyphi A infections clustering nearby one another in low lying areas [13,14]. Drinking water has long been recognized as an important route of *Salmonella* Typhi and Paratyphi A [13]. Municipal water in the Nepal's southern state of Bharatpur was implicated in the largest single source outbreak of typhoid fever recorded to date [15]. *Salmonella* Typhi has been isolated from municipal taps from the Kathmandu Valley on multiple occasions [10,13,16]. Work by Karkey et al. found high levels *Salmonella* Typhi (77% sample positivity) and *Salmonella* Paratyphi A (70%) DNA in water collected from public drinking water spouts [10]. In the twelve years since their samples were collected, lowering aquifer levels and a large earthquake rendered many of the aquifers they tested inoperable yet enteric fever cases in the Kathmandu Valley have remained high, indicating a potentially different dominant route of transmission in the area [3,10,17]. Mounting evidence shows *Salmonella* Paratyphi A may follow different transmission dynamics than *S.* Typhi [18,19]. Specifically, a case control study from the Kathmandu Valley showed *S.* Paratyphi A was likely transmitted by food while *Salmonella* Typhi transmission was associated with poor quality water usage [19]. A recent case-control study from Malawi also found use of river water for cooking and cleaning to be the strongest individual risk factor for enteric fever [20]. In the Kathmandu Valley, *Salmonella* Paratyphi A infection clusters are more geographically dispersed than those of *S.* Typhi clusters and are most often found in areas close to the Bagmati River [13]. Additionally, researchers studying enteric fever in Chile in the 1980's had discovered enteric fever causing bacteria in the river and irrigation water downstream of the capital, Santiago, that was being used to irrigate crops [21]. This evidence was used to shape agricultural policy, banning the growth of salad greens downstream of Santiago and ultimately led to a decrease of enteric fever incidence in the area [22]. The Kathmandu Valley, a dense urban area crisscrossed with rivers and surrounded by agriculture, could potentially exhibit a similar cycle of enteric fever transmission, but this exposure pathway has yet to be examined. This study seeks to understand the extent to which different water sources are contaminated with enteric fever causing bacteria *S.* Typhi and *S.* Paratyphi A across the urban Kathmandu Valley and more rural Kavre District of Nepal.

Diagnosis of enteric fever in humans is done through culture based methods, but culturing enteric fever causing *Salmonellas* from the environment is notoriously difficult, as the bacterium are hypothesized to enter a viable but not culturable state in surface water [10,23,24].

PCR techniques are commonly used to determine if *Salmonella* Typhi or *Salmonella* Paratyphi A DNA is present in environmental samples, but detecting DNA may not indicate the presence of viable bacteria [25–29]. For *Yersinia pestis* and *Bacillus anthracis*, researchers use a liquid culture enrichment step and duplicate DNA extractions to test for pathogen viability [30,31]. We have adapted this method for *Salmonella* Typhi and *Salmonella* Paratyphi A. We test drinking water and surface water (river) samples from both the Kathmandu and Kavre Districts and record human river interactions to understand the potential role of other waterborne transmission routes of enteric fever in this area.

## Methods

### Household drinking water sampling

For drinking water samples, we enrolled a geographically random, population-based sample from the SEAP hospitals catchment area which included urban, peri-urban and rural communities [32]. The catchment areas were split into grids and grid clusters were randomly selected to receive a healthcare utilization survey, of which 25,473 households were enrolled. We further randomly selected households with children (age 0–4 years, age 5–9 years, age 10–15 years, and age 16–25 years) for water sampling. The randomization process used has been previously published [32,33]. The locations of the households selected can be seen below (Fig 1).

At each household, team members collected a one-liter sample of the house's drinking water in a sterile Whirlpak bag (Nasco product Number: **B01027)** and secured it with bag locking pipe closures (Nasco product number: **B01595**). Samples were either collected directly from the tap or poured into a Whirlpak bag. If the water container was too heavy to lift, the sample was spooned from the container into the Whirlpak bag, using the same cup or device that the household used to transfer the water. For households receiving water from a municipal water source, we collected the sample from the actively running household pipe inlet to the house. If water was stored before drinking, we collected the sample from its primary storage

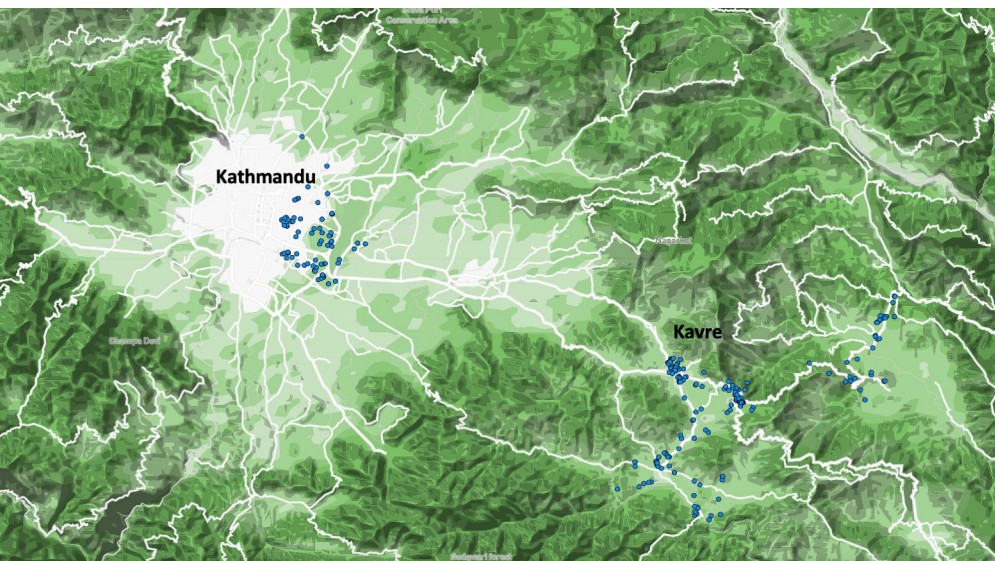

**Fig 1. Drinking water sampling locations, Kathmandu Valley and Kavre district, February to October 2019.** The location of the households from which our team collected the 370 drinking water samples reported on in this study. The aggregation of clusters of points in Kavre reflects the two peri-urban areas of the region, Banepa to the left and Dhulikhel, to the right.

container (i.e. underground or above building water tanks). In cases where primary storage was inaccessible, we collected the sample from the family's backup water reserve. For households using water from communal spigots, we collected water directly from the source spigot. If the communal spigot was not functioning, we collected a sample from a storage tank in the house, if the storage tank was inaccessible, sample was collected from water stored in a jug. For households purchasing water from a private vendor, in bottles, jars or tanks, we collected the sample directly from bottles or jars or primary storage jugs or tanks. All samples were put on ice for transportation back to the laboratory. At each household questionnaires were collected on household drinking water sourcing, water purification processes using the Redcap Mobile application v5.19.15 (Vanderbilt University, Nashville, TN) on password protected tablets. Protocol available at (DOI: dx.doi.org/10.17504/protocols.io.ewov1o6bplr2/v1, accessed 09/25/2023)

## Household drinking water assay

We performed a series of limit of detection and spiking experiments to characterize the sensitivity of the laboratory protocols we used for the drinking water assay. More information on these experiments and their results can be found in S1 Text.

When drinking water samples were collected, we stored samples at 4˚C and processed them within 24 hours of collection. Processing involved filtering the sample through a 0.45μm cellulose nitrate filter in a 250ml Nalgene Analytical Test Filter Funnel using a vacuum pump (Shiva Industries, New Delhi, India). Using sterile forceps and scissors, we cut the filter membrane into 4 even pieces and inserted them into a 5ml Power Water DNA Bead Tube with the sample side facing inward. We performed DNA extraction using a Qiagen Power Water Kit (DOI: dx.doi.org/10.17504/protocols.io.261ge3mdyl47/v1, access date 7/19/2023). After extraction, we performed real-time PCR (RT-PCR) to detect *Salmonella* Typhi and *Salmonella* Paratyphi A DNA using previously published primers and probes from Nga et al. [34]. RT-PCR reactions were performed in 20 μl reaction volumes consisting of 10μl master mix (TaqMan Environmental Master Mix 2.0), 0.8 μl of the forward and reverse primers (10 μM concentration), 0.4 μl of the probe (10 μM concentration), 4 μL template DNA and 4μL DNase/ Rnase-Free water, for 45 cycles. We ran all samples in duplicate and included two nuclease free water PCR negative controls on each 96 well plate. We chose a cycle threshold (Ct) cutoff value of 39 for determining sample positivity, as this achieved 100% specificity with process controls.

## River water sampling

We collected monthly water samples on 20 sampling trips between November 2019 and July 2021 from 19 environmental surveillance sites (ES sites). Among them, 16 sites were spread across 4 rivers within the Kathmandu Valley. Sites in each river were located at distances 1, 5 and 10 km upstream of their confluence point with the main Bagmati river. We also sampled downstream points after the rivers combined, including one location before the main confluence point and then three downstream points after all rivers combined at 1, 5, and 10 km downstream. In the more rural Kavre, samples were collected at 3 sites, 5 km apart along the Punyamata river with the middle site being directly downstream of the city of Banepa. We chose this sampling strategy because it allowed us to sample from a range of environmental settings, including locations upstream of the major city center where settlement was relatively sparse, concentrate most sample locations within the population dense areas, and sample at locations in agricultural areas downstream of cities, Kathmandu and Banepa. At each site, we collected a single 50 mL river water sample directly into a Falcon tube, the exterior of which we then wiped down with a 0.5% bleach solution and 70% ethanol and placed into separate

bags on ice for transportation to the laboratory. Each day of collection research staff also collected a field negative control at the final sampling site by pouring sterilized distilled water from a bottle into a Falcon tube and placing it in the sample bag for transport back to the laboratory. At each sampling location and timepoint, team members used a standardized survey to record information on abiotic factors that may affect bacterial input or survival in the water. The survey included questions on including observation of sewage pipes draining into the water, fecal matter near the site, and human / animal presence and activities at the sampling point. All information was entered into the mobile application Redcap v5.19.15 (Vanderbilt University, Nashville, TN). Team members also used a probe (Apera Instruments, Columbus, OH, AI311) to collect readings of the water pH, temperature, and oxygen reduction potential. Our field sampling protocol can be found online at (DOI: dx.doi.org/10.17504/protocols.io. 8epv5j836l1b/v1, accessed 7/19/2023)

## River water assay

Upon returning to the laboratory, we vortexed samples and pipetted each into three parts. We used a densitometer (DEN-1, Grant Instruments Ltd., England) on the first 5 mL of the sample to conduct a turbidity test. The second 200μl of sample was used to detect *Escherichia. coli* as a measure of fecal contamination. We created a 1:200 dilution of this sample aliquot in sterile PBS and then plated the solution onto 3M *E.coli/* Coliform Count Plates for overnight incubation at 37˚C. Blue colonies with gas present the next day were recorded as *E. coli* coliforms.

We processed the remaining 44.8 mL of sample using differential centrifugation and enrichment PCR. We used different sample processing, extraction and assaying methods for the river water, compared with the drinking water samples, due to high levels of particulate and sewage contamination river water samples and the low PCR positivity observed in the drinking water samples taken from households. First, we centrifuged the sample at 2000 rpm for 1 minute to separate larger debris. We transferred the supernatant to a new sterile tube and then centrifuged it at 4000 rpm for an additional 25 minutes to obtain the bacterial pellet. We resuspended the pellet in 0.5ml of sterile distilled water and then pipetted it into 10ml of Selenite F enrichment media. We pipetted one mL of the Selenite F bacterial mixture out of the culture media before incubation and performed DNA extraction using the Qiagen Blood and Tissue Kit manufacturer specifications. This sample was considered the time 0h extraction. The remaining 9.5mL of Selenite F broth was then incubated at 37˚C for 16 hours. At the 16-hour time point, we performed DNA extraction on an additional 1mL of this liquid media following the same procedures, which we called the time 16h extraction. We stored DNA extracts at -20˚C until further processing. We additionally incubated and processed an uninoculated Selenite F media control each day as a laboratory control.

We performed real-time PCR on the time 0h and time 16h extractions, in duplicate, using the same reaction volumes and conditions specifications described for the drinking water assay section (2.2), except the amplification for these samples was performed for 40 cycles instead of 45. Samples were considered positive if either PCR duplicate from the time 16h extraction had a Ct (cycle threshold) value of $\leq 35$. While other environmental surveillance studies have used Ct value cutoffs of up 38 or 39, we chose to use this more conservative measure to account for possible nonspecific binding of the primers to Salmonella Typhi and to ensure spurious results were not considered positive [30,31]. We defined samples as potentially viable when the PCR results showed a minimum of 3 Ct downwards shift between extractions done at time 0 and time 16 and if the sample had a Ct value of $< 35$ at the endpoint (time 16h) extraction [30,31]. The river water laboratory processes can be found at (DOI: dx.doi.org/10. 17504/protocols.io.261ge3mxol47/v1, access date 7/19/2023)

## Statistical analyses

We assembled maps of households from which drinking water was collected and river sample points that tested positive in R using a basemap created in Mapbox Studio [35,36]. We assessed differences in drinking water sourcing between Kathmandu and Kavre using chi square tests and p<0.05 as a measure of statistical significance. For Fig 2, We obtained data on daily rainfall from weather station data obtained through the Nepal Department of Hydrology and Meteorology and plotted in R using *ggplot2* [25,26]. We used locally weighted regression (loess) to create smooth lines of rainfall by month (Fig 2B) and to relate river water positivity and month of the year (2C). For water sample detection (2C), we created separate smooth lines for January–July and September–December due to there being a data gap in August.

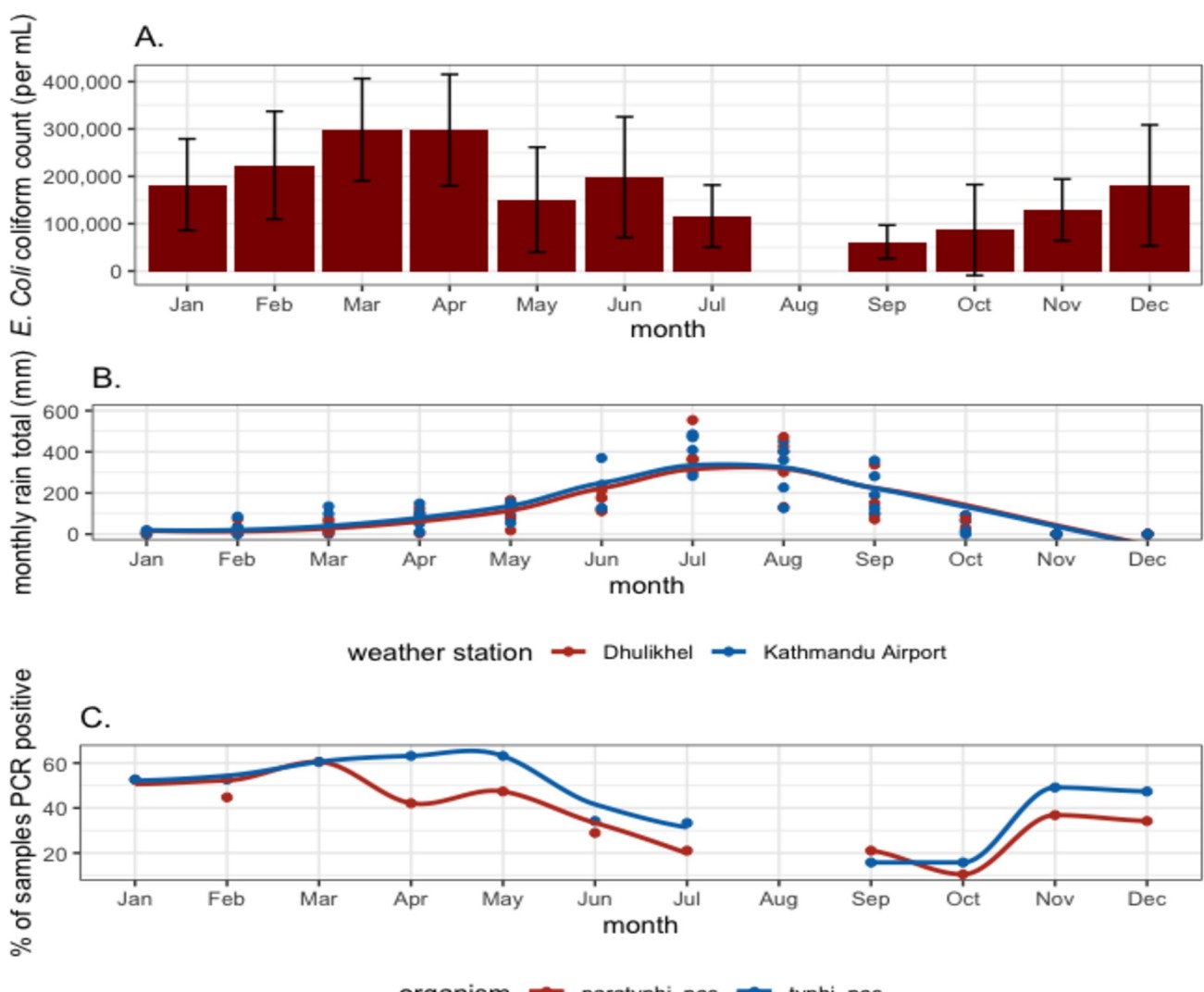

**Fig 2. Presence of typhoidal *Salmonella* DNA and *E. coli* concentration and rainfall by month in the Kathmandu and Kavre Districts.** A. The mean number of *E. coli* coliforms present per mL of sample by month from July 2020 to July 2021. There were 6 coliform samples collected in each month except for July when coliform samples were taken on the first and third week of the month for a total of 11 samples. August was missed in both years of sampling due to COVID-19 lockdowns. B. Monthly rainfall from 2015–2019 taken from Kathmandu Airport and Kavre government meteorological stations (n = 5 for each location in each month plotted). C. The proportion of river water samples PCR positive for *Salmonella* Typhi or Paratyphi A by month in 380 samples from July 2019 to May 2021.

We constructed generalized linear mixed effect models, employing the function glmer() from the "lme4" package in R to test for differences between sample positivity based on seasonality (monsoon vs. dry), region (Kathmandu vs. Kavre), and segments of the Kathmandu Valley rivers (upstream, city center, and downstream). Sample site id was included as a random effect in each model. Statistical significance was assessed at a 95% level using a Wald test.

We used a standardized protocol to estimate the hydrological catchments for each river sampling point using the Arc Hydro extension in ArcMap 10.7 (ESRI, Redlands, California, USA) [37]. Catchments were then geospatially intersected with the WorldPop gridded population dataset for the reference year 2020 to estimate total upstream catchment population at each sampling location [28]. We modeled the proportion of samples, collected from the Kathmandu region, as a function of distance from the main river confluence point using a generalized additive model with logistic link function (R package "mcgv") [29]. We conducted all statistical analyses using R (version 4.0.4). The code used for cleaning the data and generating visualizations found in this text can be found at (https://github.com/chrisleboa/nepal_typhoid, accessed 7/19/2023) Datasets including those used to make Figs are located on Dryad (DOI: 10.6078/D1ZD84, accessed 8/30/2023) [38].

## Results

### Household drinking water testing

Three hundred and seventy drinking water samples were collected from households between February and October 2019. (146 in Kathmandu and 224 in Kavre) (Fig 1). Table 1 outlines the characteristics of the samples collected and the samples that tested positive.

Households' main source of drinking water varied: 154 samples (42%) came from the municipal water supply, 84 (23%) of the samples came from jugs purchased from private companies, 49 samples (13%) had an unknown origin, 41 samples (11%) came from surface water, 35 (9.5%) came from pipes of unknown origin, 5 samples (1.4%) were from a well at the house, and 2 samples (0.5%) came from water dropped off by a water tanker truck. Households in the Kavre region were more likely to get water from the municipal water supply than those in Kathmandu (108/224 vs 46/146 p = 0.0014) and were more likely to use surface water for drinking purposes (34/224 vs 7/146 p = 0.0019) while households in Kathmandu were more likely to purchase their drinking water from private companies, both jug and tanker samples (63/146 vs 21/224 p < 0.01). 77% of samples from pipes of unknown origin came from the more rural Kavre region. 38% of households that responded (n = 289/370) reported treating their water before drinking it. Three samples that tested positive for *Salmonella* Typhi came

**Table 1. Household drinking water results, Kathmandu Valley and Kavre districts, February to October 2019.**

| Sample Type | Kathmandu | Kavre | Combined | Typhi Positive | Paratyphi Positive |
|---|---|---|---|---|---|
| Municipal | 46 | 108 | 154 | 4 | 1 |
| Pipe | 8 | 27 | 35 | 0 | 0 |
| Private company | 63 | 21 | 84 | 1 | 0 |
| Surface water | 7 | 34 | 41 | 1 | 0 |
| Tanker | 1 | 1 | 2 | 0 | 0 |
| Well | 3 | 2 | 5 | 0 | 1 |
| NA | 18 | 31 | 49 | 1 | 2 |
| Total | 146 | 224 | 370 | 7 | 4 |

The above table outlines the types and number of drinking water samples collected from households found to contain *Salmonella* Typhi and *Salmonella* Paratyphi A.

from households that reported treating their water, two from households that did not treat water and two from households that had not answered the question. Of the 7 samples that tested positive for *Salmonella* Typhi, one had come from Kathmandu while the other 6 originated from households in the Kavre region. Four of the positive samples came from municipal water (including the sample from Kathmandu, one came from a water sample from a private company, one came from surface water, and the last was undefined. Of the four samples that tested positive for *Salmonella* Paratyphi, three came from Kathmandu and a single sample came from Kavre. Two of the samples' origin was undefined, one came from a well and one came from the municipal supply.

## River water testing

We collected 380 water samples from the 19 sampling points, each site sampled 20 times between November 2019 and July 2021, on approximately a monthly basis, with brief interruptions due to COVID-19 associated lockdowns. Table 2 shows the distribution of river water sample detection by seasonality and space.

45% (n = 171) of river water samples tested positive for *Salmonella* Typhi DNA and 40% (n = 152) of the samples tested positive for *Salmonella* Paratyphi A at the time 16-hour extraction. Overall positivity was much lower before the enrichment step, with only 13 samples testing positive for *Salmonella* Typhi (3.4%) and three samples from *Salmonella* Paratyphi (0.7%). Of samples that tested positive at endline, 92% (158/171) of the *Salmonella* Typhi detected samples showed a > 3 CT shift between baseline and endline extractions while 95% (145/152) of *Salmonella* Paratyphi samples showed this shift, suggesting viability. Of all samples run, 42% (158/380) showed viability (a reduction of 3 Ct between time 0 and 16 extractions) for *Salmonella* Typhi and 38% (145/380) showed potential viability for *Salmonella* Paratyphi. We did not observe any association between pH, water temperature, or oxygen reduction potential, with these values changing irrespective of the detection rate at a location.

## Seasonal variability

Sample positivity for typhoidal *Salmonellas* varied by month, with a lower percentage of locations testing positive during monsoon months (June—September) compared to the other parts of the year (*Salmonella* Typhi 31% in monsoon, 51% non-monsoon p = <0.01, *Salmonella* Paratyphi 26% monsoon, 46% non-monsoon p < 0.01) (Fig 2). We observed a similar reduction in *E. coli* concentration during monsoon months.

**Table 2. Results of river water testing for *Salmonella* Typhi and *Salmonella* Paratyphi A in the Kathmandu and Kavre Districts, November 2019 to July 2021.**

| | | Number Tested | Typhi Pos | Typhi Viable | Paratyphi Pos | Paratyphi Viable | Typhi % Positive | Paratyphi % Positive |
|---|---|---|---|---|---|---|---|---|
| Seasonality | Monsoon season | 266 | 136 | 125 | 122 | 115 | 51 | 46 |
| | Dry season | 114 | 35 | 33 | 30 | 30 | 31 | 26 |
| District | Kathmandu | 320 | 156 | 143 | 135 | 129 | 49 | 42 |
| | Kavre | 60 | 15 | 15 | 17 | 16 | 25 | 28 |
| River Segment | City center | 180 | 107 | 99 | 94 | 91 | 59 | 52 |
| | Downstream | 60 | 39 | 34 | 36 | 34 | 65 | 60 |
| | Upstream | 80 | 10 | 10 | 5 | 4 | 13 | 6.3 |
| Total | Overall | 380 | 171 | 158 | 152 | 145 | 45 | 40 |

This table describes the numbers of samples tested by season, the number of samples which tested positive, potentially viable, and the percentage of samples that tested positive.

## Spatial variability

River water contamination with typhoidal *Salmonellas* varied spatially. *Salmonella* Typhi was detected at 10 sampling sites more than 10 times, representing over 50% of sampling trips while *Salmonella* Paratyphi was detected at 9 sites on at least 10 occasions (Fig 3) Three

### A. *S.* Typhi

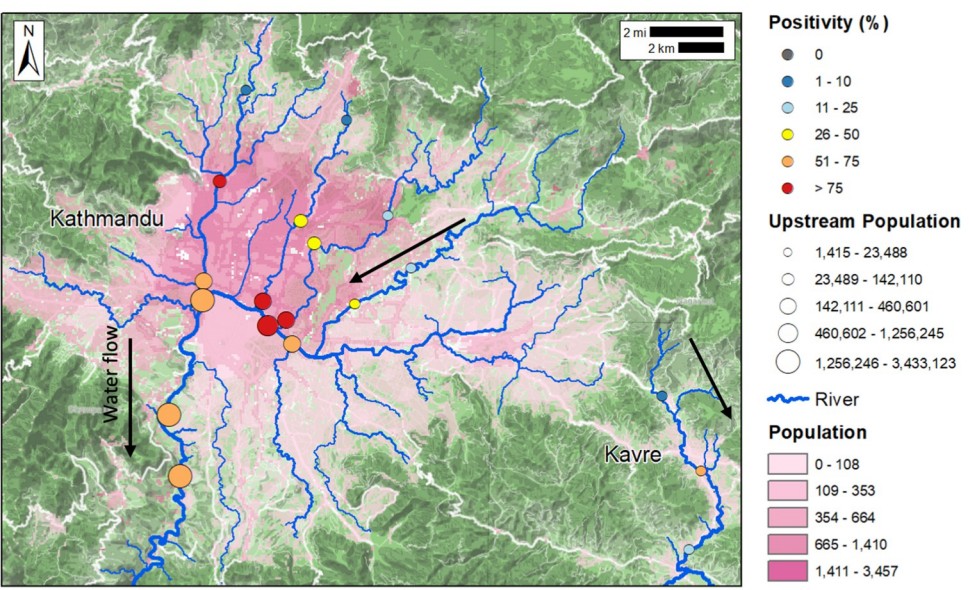

### B. *S.* Paratyphi A

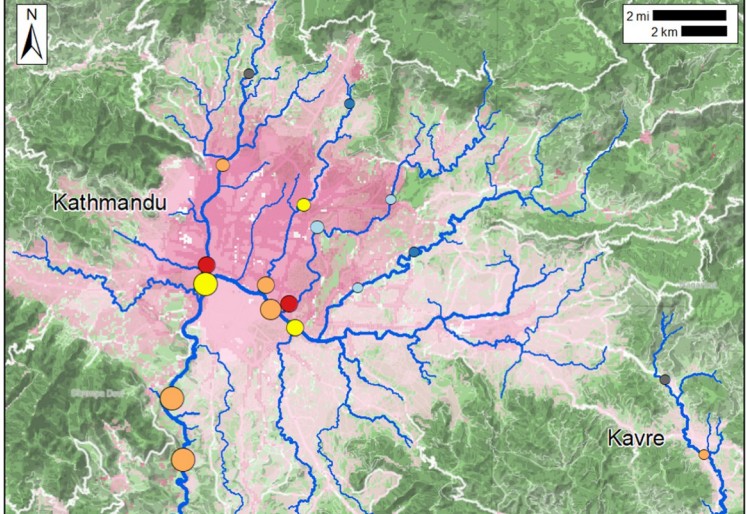

**Fig 3. Frequency of detection of *Salmonella* Typhi and *Salmonella* Paratyphi A in river water in Kathmandu and Kavre Districts between November 2019 and July 2021.** The 19 sampling points were each sampled 20 times at approximately a monthly interval. The proportion positive for *Salmonella* Typhi (grid A) and *Salmonella* Paratyphi A (grid B) are indicated by color of the sampling point. The size of the sampling point indicates the number of people estimated to live upstream of that point which was calculated using ArcGIS Hydrosheds. The population density in units of people per 100m$^2$ is indicated in pink shading. Water flow direction goes from North to South as indicated by the black arrows in the upper pane.

sampling locations tested positive for *Salmonella* Paratyphi once; two sites tested positive a single time for *Salmonella* Typhi. Central Kathmandu (samples 1km and 5 km upstream of confluence) was highly contaminated with 59% (107/180) of samples testing positive for *Salmonella* Typhi and 52% (94/180) positive for *Salmonella* Paratyphi. Samples collected downstream of the city were positive 65% (39/60) of the time for *Salmonella* Typhi and 60% (36/60) for *Salmonella* Paratyphi A. Areas upstream from densely populated areas (samples 10 km upstream from the confluence) had significantly lower positivity rates 13% (10/80) for *Salmonella* Typhi; 6.3% (5/80) for *Salmonella* Paratyphi A. Our mixed effects model shows that the odds of a sample from the city center containing *Salmonella* Typhi DNA was 11.3 times more likely (95% CI: 4.3, 33.1) and from a downstream site 14.5 times more likely (95% CI: 4.4–50.8) than the odds of an upstream sample testing positive for *Salmonella* Typhi. A generalized additive model (S1 Fig) also shows increasing rates of positivity the further downstream samples are taken, except for a slight tapering of sample positivity for *S.* Typhi at the point 10 km downstream of Kathmandu city limits.

Samples collected from Kathmandu Valley were slightly, but not significantly more often positive than those collected from the more rural Kavre region. For *Salmonella* Typhi, 49% of Kathmandu Valley samples were positive (156/320) while 25% (15/60) of Kavre samples were positive (p = 0.113). *Salmonella* Paratyphi showed a similar pattern with 42% (135/320) and 28% (17/60) of samples testing positive in Kathmandu and Kavre respectively (p = 0.357).

## Observations of river usage

At fifteen of the nineteen sampling locations, there were sewage or rainwater runoff pipes entering the river system (13 from Kathmandu and 2 from Kavre). At 8 of 19 of these sites, we observed untreated sewage actively draining into the water. There were agricultural fields along the banks of the river at 7 of 19 of the sampling sites, including at the two most downstream sampling locations from the Kathmandu Valley. We observed an irrigation pipe leading from the river to a nearby field at one sampling location. Our field team also observed people interacting with the rivers, as recorded in Table 3.

People walked through the rivers at 12 of 19 sampling locations, washed clothes in the rivers at 7 of 19 sampling locations, bathed in the rivers at 4 of 19 sampling locations, and washed vegetables in the rivers 2 sampling points on 12 separate occasions (Fig 4). The corresponding water sample tested positive for *Salmonella* Typhi on 9 occasions when people were observed walking, 5 occasions when people were washing clothes, 4 occasions when people were bathing and 4 occasions at which people were washing vegetables (carrots and spinach). Conversations with these individuals indicated that the vegetables were being washed before sale in markets in and around Kathmandu. Human interactions with rivers mainly occurred within the Kathmandu Valley. People were observed walking, washing clothes and bathing in only the most upstream point in the Kavre region.

**Table 3. Human interactions with river systems, Kathmandu and Kavre Districts, November 2019 to July 2021.**

| Action | Number of Times Observed | Number of Locations Observed | Action when pos. for *Salmonella* Typhi | Action when pos. for *Salmonella* Paratyphi |
|---|---|---|---|---|
| Washing Vegetables | 12 | 2 | 4 | 2 |
| Walking across river | 25 | 12 | 9 | 6 |
| Washing clothes in river | 21 | 7 | 5 | 4 |
| Bathing in river | 11 | 4 | 4 | 2 |

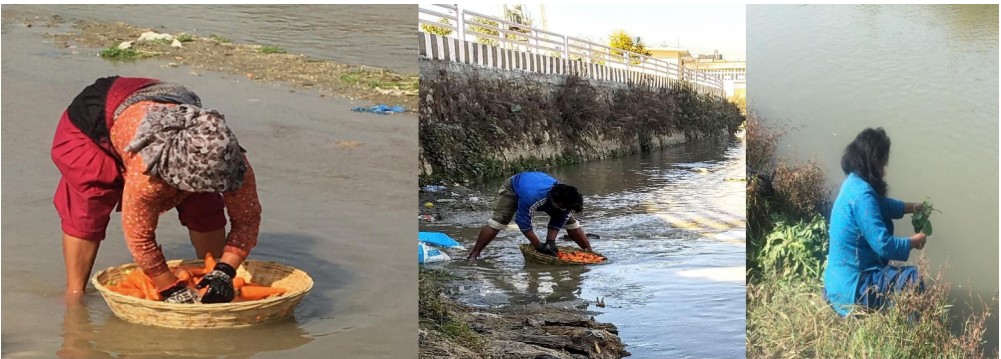

**Fig 4. Observations of individuals washing vegetables in rivers of Kathmandu: At two sampling points across twelve instances, individuals were observed washing vegetables from the market.** These images show carrots and spinach, the most common vegetables observed being washed.

## Discussion

### Household drinking water and fecal-oral transmission

In a population-based sample of household drinking water, we found low levels of contamination with typhoid fever causing *Salmonellas*. By contrast, sampling of river waters revealed the presence of *Salmonella* Typhi and *Salmonella* Paratyphi A in over 40% of river water samples. Contamination with typhoidal *Salmonellas* occurred more frequently in river segments within and downstream of densely populated areas. Additionally, the use of sewage-containing river water for washing vegetables, and the potential uses of irrigation of nearby agricultural fields, suggests possible routes for exposure or the food-borne spread of typhoidal *Salmonellas*.

Our findings diverge from previous studies in South Asia, which have reported high rates of *Salmonella* Typhi in drinking water and had suggested drinking water as the primary route for bacteria exposure [10,39]. The Karkey et al. study in Lalitpur, an area within the Kathmandu Valley, found 77% of the samples collected from 10 stone spouts contained *Salmonella* Typhi DNA [9]. When we sampled drinking water from households in the Kathmandu Valley and Kavre Districts, we found no households reported using these spouts for drinking water procurement. Many of these spouts have become inoperable due to groundwater over extraction (the aquifer below Kathmandu is falling at a rate of 2 meters per year) and earthquake damage [17,40–42]. In our study, less than 50% of households sampled from Kathmandu or the Kavre region (154/370) used municipal water as their main drinking water source. Instead, private water companies (63/146) supplied the drinking water for many of the households surveyed. Kathmandu suffers from increasing levels of municipal water scarcity, with demand (360,000 m$^3$/day) exceeding supply (ranging between 90,000 and 150,000 m$^3$/day) [40]. An analysis by Shrestha et al. showed that after the infrastructure damage of the 2015 Gorka earthquake, households decreased municipal water usage and increased the use of tanker water from private companies [43–45]. The sources for private company water may be from upstream or less populated areas and may contain less sewage in than water originating in or being piped through the Kathmandu Valley. In Kavre, a significant portion of households used surface water or locally constructed pipe systems for their drinking water (54/224). As this region is not downstream from a large population center, these surface samples were less likely to be contaminated with sewage effluent before reaching these rural and semi-rural households. Of the 11 drinking water samples that tested positive for *Salmonella* Typhi or Paratyphi A, 5 of the positive samples (45%) came from the municipal water supply, which due to the intermittent flow allows for surrounding sewage to leech into the pipes [46,47].

## Surface water as a foci for transmission

River water yielded high rates of positivity (*Salmonella* Typhi 45% and *Salmonella* Paratyphi 42%), particularly in segments located within and downstream densely populated areas. These findings indicate fecal matter from population centers is entering river systems, and that the rivers are a potential hazard to human health even many kilometers past city boundaries. In Kathmandu, municipal sewage systems are absent or incomplete [48]. Our findings augment similar work describing high levels of fecal coliforms in the waters in Kathmandu's river system [49]. Two of the three sampling sites that most often tested positive, Bagmati 1 (75% positivity rate for *Salmonella* Typhi and *Salmonella* Paratyphi A, and Combination 0 (75% for *Salmonella* Typhi and 70% *Salmonella* Paratyphi A) are spatially adjacent to the high incidence enteric fever cluster along the Bagmati river, described by Baker et al. [13]. The river samples adding further evidence that this area continues to be a hotspot of enteric fever transmission. Our findings indicate that the samples from downstream river locations, adjacent to agricultural fields, were positive for *Salmonella* Typhi 65% of the time, suggesting that nearby irrigation water may also be contaminated and used on crops. This potential pathway of transmission would mirror what was described in Chile in the 1980's [21,22]. Additional analysis of irrigation water in areas downstream of Kathmandu could indicate whether irrigation of crops with *Salmonella* Typhi contaminated water presents a potential foci of transmission. Our structured observations added evidence that rivers may be at locations at which people are exposed to typhoidal *Salmonellas*, as we observed agricultural workers washing carrots and spinach in the rivers before the market. Our work provides evidence that river water in Kathmandu is heavily contaminated with *Salmonella* Typhi and Paratyphi A, including several kilometers downstream of the population center and in some cases, people directly contact rivers or use river water for washing products. However, the scale to which this contact occurs is still not well characterized. The positivity of river water samples could simply be evidence of shedding from infected individuals and not a primary source of risk for disease exposure among the general population. Additional epidemiological investigations could be used to better understand the pathways of transmission in this context.

## Surface water seasonality

We found both *Salmonella* Typhi and *Salmonella* Paratyphi A contamination across the greatest number of sample sites during the dry season (October—May), we also found higher levels of *E. coli* contamination during these months. The higher levels of *Salmonella* Typhi contamination during these months contrasts the seasonal pattern of enteric fever incidence in the Kathmandu Valley, which have been reported to peak during the monsoon season [10,50]. We hypothesize the river may have more detectable levels of *Salmonella* Typhi during dry seasons due to dilutional effects of monsoon rains increasing the overall water flow through the valley's river systems while the levels of sewage generated remain constant.

## Environmental sampling and case data

We found sample locations from the Kathmandu Valley were slightly but not statistically significantly more often contaminated with *Salmonella* Typhi and *Salmonella* Paratyphi A (49% *S.* Typhi and 42% *S.* Paratyphi A) than locations from the more rural Kavre District (25% *S.* Typhi and 28% *S.* Paratyphi). Sero-epidemiological data from these two regions similarly shows higher levels of *S.* Typhi exposure in Kathmandu (6.6 per 100 person years) compared to Kavre (5.8 per 100 person years) [4]. Environmental sampling for pathogens can be used to not only show sources of transmission but also can be developed into as a sentinel surveillance tool to understand ongoing nearby disease transmission. Wastewater based epidemiology has

been used in the COVID-19 pandemic to understand trends in otherwise unreported cases, but more work must be done to relate a positive environmental sample with associated disease incidence in nearby populations [11,51,52].

## Limitations and next steps

The results of this work must be interpreted in the context of numerous limitations. First, drinking water samples were analyzed without an enrichment step while river water samples were extracted both pre and post enrichment (the time 0h and time 16h samples). If an enrichment step was also added to drinking water samples, they may have more often shown detectable levels of *Salmonella* Typhi and *Salmonella* Paratyphi A. The earlier work by Karkey et. al also did not include an enrichment step before quantification through qPCR [10]. We attempted to culture and isolate *Salmonella* Typhi from the water samples to directly demonstrate the viability of the organism in these samples, but were unsuccessful in doing so due to growth of other environmental bacterium, as was the experience of earlier work trying to culture *Salmonella* Typhi from water [53]. To overcome this limitation, we used enrichment PCR in the river water samples, as previously described by the EPA for use on *Yersinia pestis* and other bioterrorism agents [54]. With this technique, a 2-cycle threshold shift represents a 4-fold increase in DNA detected, which is greater than would be expected by technical variability, providing evidence of bacterial growth but it is a proxy for the number of colony forming units present [27,45,55]. A positive river water sample test gives a binary, crude result indicating circulating disease nearby, but it is not possible to identify where the bacterium originated. Additionally, the COVID-19 pandemic spurred lifestyle changes in the Kathmandu Valley, possibly shifting exposure pathways as well as overall enteric fever incidence during this study period. Additional months of data collection could be used to see if the same trends identified in this publication are reproduced independent of potential influences of the pandemic.

Despite the presence *of Salmonella* Typhi and Paratyphi A DNA contamination in samples from sites with human interactions and vegetable washing, our study does not provide definitive causal evidence of association of contaminated river water or produce with human enteric fever infections. Sampling of food at the point of consumption, and epidemiological studies assessing exposure to potentially contaminated food sources and risk of enteric fever, should be undertaken to better investigate this potential pathway. Moreover, testing irrigation water in fields alongside the contaminated sections of river could help us understand if contaminated surface waters are interacting with nearby fields. The transmission pathways of *Salmonella* Typhi and *Salmonella* Paratyphi A are complex, subject to change and multifaceted. Continuing environmental surveillance after the new typhoid vaccine introduction to Nepal could additionally help understand if the vaccine influences the prevalence of enteric fever causing *Salmonellas* in downstream water supplies.

## Conclusion

In population-based sampling of drinking water taken from households in the Kathmandu Valley, we found low abundance of typhoidal *Salmonella* DNA. Drinking water samples collected from households that tested positive came from both the Kathmandu and the Kavre districts and came from municipal pipes as well as from a private jug sample, surface water sample and from a well. We found that rivers flowing through Kathmandu were often highly contaminated with viable *Salmonella* Typhi and *Salmonella* Paratyphi A. These findings, together with observations of human use of river water for vegetable washing, suggest rivers as a potential transmission focus for enteric fever in the Kathmandu Valley. Fecal contamination of river water should be addressed through improved sewage and sanitation infrastructure,

and while such efforts are planned, they have yet to be implemented. A deeper understanding of the transmission pathways of enteric fever in this region would frame effective measures for controlling its spread. These results show other sources beyond drinking water should be considered as possible pathways for transmission in this geographical context.

## Supporting information

**S1 Fig. Proportion of samples positive for *S*. Typhi according to distance from the river confluence in central Kathmandu.**
(DOCX)

**S1 Text. Limit of detection for drinking water assay.**
(DOCX)

## Acknowledgments

A large team of individuals from multiple institutions helped make this project possible. Puspa Raj Bhatt, Bipin Thapa, Anil Khanal, Karuna Timilsina, Rojina Bhaila Shrestha, Natasha Shrestha, Shisir Ranjit, Lokmani Bhatt, Ashmita Karmacharya, Apekshya Chaulagain, Sudan Maharjan, Sabin Bikram Shahi, Sudichhya Tamrakar, Aastha Shrestha, Laxmi Chauguthi, Aarjya Tara Bajracharya, Suraj Jakibanzar, Melina Thapa and Neeru Suwal have helped with Sample/Data collection.

## Author Contributions

**Conceptualization:** Christopher LeBoa, Sneha Shrestha, Alexander T. Yu, Nishan Katuwal, Kristen Aiemjoy, Stephen P. Luby, Jason R. Andrews, Dipesh Tamrakar.

**Data curation:** Christopher LeBoa, Sneha Shrestha, Shiva Ram Naga, Alexander T. Yu, Krista Vaidya, Kristen Aiemjoy, Jason R. Andrews.

**Formal analysis:** Christopher LeBoa, Alexander T. Yu.

**Funding acquisition:** Alexander T. Yu, Kristen Aiemjoy, Denise O. Garrett, Stephen P. Luby, Jason R. Andrews.

**Investigation:** Christopher LeBoa, Jivan Shakya, Shiva Ram Naga, Sony Shrestha, Mudita Shakya, Jason R. Andrews.

**Methodology:** Christopher LeBoa, Sneha Shrestha, Jivan Shakya, Sony Shrestha, Alexander T. Yu, Jason R. Andrews.

**Project administration:** Christopher LeBoa, Sneha Shrestha, Jivan Shakya, Shiva Ram Naga, Kristen Aiemjoy, Jason R. Andrews.

**Resources:** Christopher LeBoa, Krista Vaidya, Kristen Aiemjoy, Isaac I. Bogoch.

**Software:** Christopher LeBoa, Alexander T. Yu, Christopher B. Uzzell.

**Supervision:** Rajeev Shrestha, Nishan Katuwal, Stephen P. Luby, Jason R. Andrews, Dipesh Tamrakar.

**Validation:** Christopher LeBoa, Sneha Shrestha, Jivan Shakya, Sony Shrestha.

**Visualization:** Christopher LeBoa, Christopher B. Uzzell, Jason R. Andrews.

**Writing – original draft:** Christopher LeBoa, Sneha Shrestha, Jason R. Andrews.

**Writing – review & editing:** Christopher LeBoa, Sneha Shrestha, Jivan Shakya, Shiva Ram Naga, Sony Shrestha, Mudita Shakya, Rajeev Shrestha, Krista Vaidya, Nishan Katuwal, Kristen Aiemjoy, Isaac I. Bogoch, Christopher B. Uzzell, Denise O. Garrett, Stephen P. Luby, Jason R. Andrews, Dipesh Tamrakar.

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
