## [Decision Letter · Decision Letter 0]

23 Jun 2023

Dear Mr. LeBoa,

Thank you very much for submitting your manuscript "Environmental surveillance for typhoidal Salmonellas in household and surface waters in Nepal identifies potential transmission pathways" for consideration at PLOS Neglected Tropical Diseases. As with all papers reviewed by the journal, your manuscript was reviewed by members of the editorial board and by several independent reviewers. In light of the reviews (below this email), we would like to invite the resubmission of a significantly-revised version that takes into account the reviewers' comments. 

We cannot make any decision about publication until we have seen the revised manuscript and your response to the reviewers' comments. Your revised manuscript is also likely to be sent to reviewers for further evaluation.

Sincerely,

Epco Hasker

Academic Editor

Stuart Blacksell

Section Editor

Reviewer's Responses to Questions

**Key Review Criteria Required for Acceptance?**

**Methods**

-Are the objectives of the study clearly articulated with a clear testable hypothesis stated?

-Is the study design appropriate to address the stated objectives?

-Is the population clearly described and appropriate for the hypothesis being tested?

-Is the sample size sufficient to ensure adequate power to address the hypothesis being tested?

-Were correct statistical analysis used to support conclusions?

-Are there concerns about ethical or regulatory requirements being met?

Reviewer #1: The overall objective of the study is stated as: “This study seeks to understand the extent to which different water sources are contaminated with typhoid fever causing bacteria S.Typhi and .Paratyphi A across the urban Kathmandu Valley and more rural Kavre district of Nepal.”

Very detailed methodology for water sample collection and analysis of S. Typhi and S. Paratyphi in samples is presented in the manuscript, including sensitivity testing of methods using limit of detection assay and spiking experiments. The methodology often refers to additional procedures available at different web addresses (line 182/3, line 191, line 220, line 256, line 271). 

The sampling frame used for the drinking water sampling is unclear. The authors indicate that (lines 142 – 144): “We leveraged a population-based household survey conducted between 2017 and 2019 in Kathmandu and Kavrepalanchok districts, which included urban, peri-urban and rural communities[18]. From that survey, we randomly sampled households for drinking water.” The sampling process for the original survey should be elaborated on, along with the number of households which were sampled for the current study (and how this sample size was calculated) and the method of random sampling used. 

The sampling strategy for monthly river water samples appears to included 19 sites across at least 9 locations (indicated in lines 203 – 205 as being located at distances 1, 5 and 10 km upstream and downstream from the main river confluence point in the Kathmandu Valley and at three locations 5 km apart on the Punyamata river in the more rural Kavre District). The authors indicated that they chose this sampling strategy because it allowed them to “sample from a range of environmental settings”. In the statistical methods the authors indicated they modelled the proportion of samples as a function of distance from the river confluence point using generalized additive models with logistic link functions (R package “mcgv”) (lines 267-268). The number of samples collected at each location or environmental setting is not clear making it difficult to determine if valid conclusions that can be drawn from any spatial modelling.

The authors have used a observational study of river usage at each 19 river water sampling sites to describe some of the possible sources of river water contamination. Although it is indicated that a mobile application was used to collected data on abiotic factors (line 215-216), the frequency of collection of this data at each site is not indicated. It is also not clear whether a standardised tool was used, and whether staff were trained in data collection to minimise bias.

Methods on statistical analyses (line 258-271) should be revised include:

- details of basic analysis to determine differences in the frequency of S. Typhi and S. Paratyphi sample positivity, including statistical tests and the level of statistical significance used.

- Detail of how trend lines were fitted to Figures 2B and 2C

Reviewer #2: The objectives are clear

The study design is appropriate

The population is clearly described

The sample size is sufficient

There are no ethical concerns

Reviewer #3: (No Response)

**Results**

-Does the analysis presented match the analysis plan?

-Are the results clearly and completely presented?

-Are the figures (Tables, Images) of sufficient quality for clarity?

Reviewer #1: The results section begins by presenting the findings of the limit of detection tests (lines 273-281). The relevance of these results to the overall study hypothesis is not clear. It is suggested that this is included as supplementary material.

The authors variably stratify the results for river water sample analysis by month, season, location (Kathmandu Valley and Kavre), sampling site, distance from confluence or by population density, as well as describing overall frequency of serovar detection in all samples. Drinking water sample results describes the source of household water supply and the number of sample where serovars were detected by location (Kathmandu Valley and Kavre) and water source. In both instances, data is presented in the text as counts by group (sometimes with a denominator, proportions and p values for tests of significance). The clarity for the reader of the data in this results section would be greatly improved through the presentation of data in a table/s. The text of the results section can then be reviewed and minimised by referring to these data tables.

River water testing data is somewhat difficult to interpret as it includes both outcomes of methodological validation (the role of the enrichment step and the discussion of pathogen viability) as well as aggregate and stratified water contamination data. It is suggested that the authors use subheadings and present the data tables to help the reader navigate. It is unclear how important pathogen viability is to discuss in the context of study hypothesis. 

A number of approaches are detailed in the methodology for which no results are presented in the main manuscript text. The authors are recommended to correct the mismatch in this content:

- Lines 160 – 163: household questionnaires on drinking water purification 

- Lines 214 – 220: collection of abiotic factors that may affect bacterial input or survival (water pH, temperature, oxygen reduction potential)

- Lines 267 – 268: Modelling of the proportion of samples as a function of distance from the river confluence point 

No reference has been made in the main results text of the manuscript to the data presented the supplementary information annex.

Figure titles should be amended to include details of place and time of the study, and sample size where relevant (eg Figure 1: Drinking Water Sampling Locations, Kathmandu Valley and Kavre district, February to October 2019.)

Figure 1 is unclear as includes labels for Kathmandu, Dhulikhel and Banepa, while the figure caption refers to Kavrepalanchok. The authors should avoid to include methodology of sampling in the figure caption in the results section. It is suggested that Figure 1 could be moved instead to the method section.

The title and caption for Figure 2 should include a sample size for each month as this relates to monthly rain (B) and river water sampling (C). The caption for Figure 2A should describe what statistic is represented by the error bars in this figure. As river water samples were not collected in August, the trend line provided in part C, should have a break at this point.

Figure 3 would benefit from presentation in a large format and better markers on each map for distance and orientation of upstream and downstream locations.

The authors should consider including data on observations on river usage (line 362) in a table, possibly stratified by observations at collection sites in Kathmandu Valley and Kavre if this is relevant. This data is limited in that it can only give an impression of sources of contamination – the authors should avoid to give this data too much emphasis.

The results should be carefully reviewed for consistency and accuracy – for example the number of household drinking water samples collected is variably reported as 370 (line 283), 400 (line 289).

Reviewer #2: The analysis matches the analysis plan

Results are clearly presented

The quality of the figures and images is of good quality and sufficient

Reviewer #3: (No Response)

**Conclusions**

-Are the conclusions supported by the data presented?

-Are the limitations of analysis clearly described?

-Do the authors discuss how these data can be helpful to advance our understanding of the topic under study?

-Is public health relevance addressed?

Reviewer #1: The presentation of the conclusions on household drinking water contamination should be reviewed. The authors mostly re-state the results. It is not clear where the main conclusions from the current study and the comparison to findings in the literature are separated. For example in the following text it is difficult to distinguish the what information is contributed by reference 33 and by the current study – lines 407-410 “In our study, less than 50% of households sampled from Kathmandu or the Kavre region (154/370) used municipal water as their main drinking water source although slightly more households used municipal water from Kavre than did in Kathmandu[33].” The authors should discuss the findings in context of literature relating to different transmission pathways for S. Typhi compared to S. Paratyphi. 

Conclusions on the role of contamination of vegetables from river water in the transmission pathway of S. Typhi and S. Paratyphi should not be overstated (line 432 – 441). Food sampling at the point of consumption and epidemiological studies to investigate the exposure to contaminated food in infected individuals provide better strength of evidence for associations.

Several of the limitations of the study discussed by the authors relate to the methodology used to analyse water samples. The authors should reflect on next steps and information which should be collected to better understand transmission of S. Typhi and S. Paratyphi. Environmental sampling should not be considered in isolation – reports of human cases of enteric fever and well designed epidemiological studies may better identify sites and risk factor for exposure and transmission.

Reviewer #2: The conclusions are supported by the data presented and the limitations are clearly highlighted. 

The authors describe the need for further investigations to link positivity of river water samples to risk of exposure to typhoidal pathogens and disease

Reviewer #3: (No Response)

**Editorial and Data Presentation Modifications?**

Reviewer #1: The focus of this study is the understanding of environmental (water) sources of Salmonella enterica serovars Typhi and Paratyphi with the view to identifying possible pathways of transmission in the Kathmandu Valley and rural Kavre district of Nepal. The author introduce the use of vaccines and antibiotic treatment and their partial effectiveness in controlling transmission, as a means of justifying why environmental sampling is also necessary (lines 97-107). The authors should consider consolidating the information on vaccines as this should be provided for context of research focus only and doesn’t relate to conclusions of the research. The authors use an example from Bangladesh to describe effectiveness of TCV mass vaccination campaign. Given the location of the research, it would be more appropriate to discuss vaccination against enteric fever in Nepal. The authors could consider the study by Shayka et al (The Lancet Global Health. 2021;9(11):e1561-1568) and acknowledge that TCV has recently been introduced in Nepal (https://www.unicef.org/nepal/press-releases/nepal-introduces-typhoid-vaccine-routine-immunisation-across-country).

The introduction of the paper only briefly focuses on the epidemiology of typhoid fever transmission in Nepal (lines 94-96, with a small mention of the impact on typhoid case numbers resulting from changes in aquifer levels and after the Kathmandu Valley earthquake (lines 121 and 123)). It is recommended that the manuscript includes more information on the geographical and temporal distribution of cases in Nepal and specifically in the Kathmandu Valley and Kavre district (for example Andrews et al The Journal of Infectious Diseases. 2017;jix221:S1-S8. and Baker S, et al. Open Biol 2011; 1:110008.) especially as these relate to the context and objectives of the current study. Data from references 9 and 39 could also be included here.

The title of this paper and the body of the manuscript, including the conclusions frequently refers to the activities of environmental water sampling and testing described in the paper as “environmental surveillance”. While river water samples were collected at designate locations approximately monthly between November 2019 and July 2021, this manuscript should be careful that it does not overstate the presented research as surveillance. A previous paper by some of the co-authors very nicely prepares a rationale for environmental surveillance as a tool for identifying typhoid transmission (Andrews et al CID, 2020:71 (Suppl 2): listed as reference 20 in this manuscript). The present manuscript would benefit from similar reflections from these co-authors. As a separate note, reference 20 (which appears on line 193) is incorrectly attributed to primer and probe sequences of Nga et al.

Drinking water samples collected as part of this study were tested for Salmonella enterica serotype Typhi and Salmonella enterica serotype Paratyphi. This manuscript refers often to “typhoid fever” or “typhoidal Salmonellas” or “typhoid” when referring to these two serotypes. The introduction and discussion of this paper makes some general statements around drinking water and “typhoid fever exposure”, dominant mechanisms of (typhoid) transmission or “Salmonella exposure” (lines 112 to 117). The authors are recommended to review the nomenclature used in their manuscript – and should take care when generalising disease due to paratyphoid and typhoid together especially as some studies suggest the risk factors for the development of enteric fever due to S. Typhi and S. Paratyphi differ (Karkey A et al PLoS Negl Trop Dis. 2013;7(8):e2391 and Levantesi, C. et al Food Res. Int., 45 (2012), pp. 587-602). 

The expertise of some of the senior co-authors of this manuscript (including Associate Professor Andrews) could be drawn on to improve the clarity around these descriptions.

Given the level of methodological detail included in this manuscript, the rationale for use of the selected sample analysis approaches for household drinking water and river water samples has not been included in the introduction, with only a small discussion of the included about the limitations associated with the use of enrichment steps (lines 469 – 481). More background should be provided on gold standard testing approaches for S. Typhi and S. Paratyphi in water samples, and why qPCR and enrichment methods were used.

Other more general edits are also recommended:

- The text is sometimes clumsy when referring to and describing other published work. For example (lines 116-121) has a long description of work which appears in reference [9]. Initial the authors mention “previous work” without a reference. The manuscript directly quotes the conclusions of Karkey and colleagues, without mentioning the authors by name (“they found”…. “which led them to conclude”). For brevity and clarity it is suggested that structure of these sentences should be reviewed. 

- Line 117 – referring to Salmonella exposure is too general: a species (Salmonella enterica) or serovar/s (S. Typhi / S. Paratyphi) must be specified.

- Refer to household drinking water or drinking water sampled from households, not “drinking water” (review throughout document)

- Kavre and Kavrepalanchok are used interchangeably – a consistent name should be used

- Notes under Figure 2 mention Coronavirus Lockdowns. It is more correct to refer to COVID-19 lockdowns or COVID-19 pandemic restrictions. 

- References to websites within the text should include an access date.

Reviewer #2: LINE 65: Please delete ‘s’ …….srepresent

LINE 90: Should be ….’over 100,000 deaths’

LINE 97 to 99: This statement is confusing .......Ty21a is a whole cell inactivated vaccine orally administered, while the Vi is a subunit vaccine intramuscularly administered

LINE 106: ‘also’ is repeated……. Ending transmission also must also

address environmental and political issues.

LINE 113: Should be ……’different times’

LINE 116: Add spacing beginning of sentence ……” In Nepal, previous work implicated…….”

LINE 186: How long were the holidays and does this affect the results?

LINE 205 to 206: Please rewrite this sentence to clearly indicate the characteristics of the locations included

LINE 218: Should ‘point’

LINE 229: To be exact would this be 44.8 mL

LINE 269: Should be…… ‘The code used…….’

LINE 277 onwards: please italicize ‘S’ for Salmonella

LINE 399: Add space beginning of new sentence

LINE 401: There is a missing word……is it ‘surveyed?’

LINE 421: There is a missing word after ‘sporadic’

LINE 423 to 424: Does it matter how further downstream a location is? Does the topography of the section of the river being sampled matter and corresponding speed of the waters going downstream?

LINE 418 to 421: for consistency you can maintain use of numbers to present figures, within the sentence.

Reviewer #3: (No Response)

**Summary and General Comments**

Reviewer #1: The overall objective of the study is stated as: “This study seeks to understand the extent to which different water sources are contaminated with typhoid fever causing bacteria S.Typhi and .Paratyphi A across the urban Kathmandu Valley and more rural Kavre district of Nepal.”

The content of this manuscript represents a large amount of work, including methodological development, however care should be taken to avoid to publish all of an academic endeavour as a single body of work while balancing with sufficient content to adequately test a hypothesis. The authors are encouraged to edit and refine the manuscript to clearly focus the presentation of the methodology and results on the main objective of the study and generate a clear narrative to arrive at the conclusions (including limitations and next steps).

Reviewer #2: I would like to commend the authors for their research and study findings, which add to the mounting evidence that the pathways leading to typhoidal disease can be influenced by not only drinking water but also by using water to wash and clean fruits and vegetables.

I have found areas in the manuscript that need to be clarified and corrected for the reader's benefit.

Reviewer #3: These study findings are very important to disseminate to prevent the transmission of S. Typhi by implementing effective preventive strategies. 

Minor comments are as follows:

It will be good if the authors can add any surveillance data to show the positivity rate of typhoid infection among people who used the contaminated water during the study time. Moreover, the authors have only observed that people wash their vegetables with river water, but they did not take any history if they have consumed these vegetables cooked or without cooking. History of taking outside foods is also important. 

The authors should mention the name of model used for the analysis in the statistical analyses section. What are the covariates used for the adjustment should be mentioned here? Have they used season as a confounding variable?

The authors should make all “S” and “salmonella” italic throughout the manuscript in case of S. Typhi and S. Paratyphi.

The authors should check the manuscript thoroughly and rewrite it where need to improve the English of this manuscript.

PLOS authors have the option to publish the peer review history of their article (what does this mean?). If published, this will include your full peer review and any attached files.

Reviewer #1: No

Reviewer #2: Yes: Dr. Chisomo Msefula

Reviewer #3: No
---

## [Editor Report · Decision Letter 1]

6 Sep 2023

Dear Mr. LeBoa,

We are pleased to inform you that your manuscript 'Environmental sampling for typhoidal Salmonellas in household and surface waters in Nepal identifies potential transmission pathways' has been provisionally accepted for publication in PLOS Neglected Tropical Diseases.

Best regards,

Epco Hasker

Academic Editor

Stuart Blacksell

Section Editor

---

## [Editor Report · Acceptance letter]

3 Oct 2023

Dear Mr. LeBoa,

We are delighted to inform you that your manuscript, "Environmental sampling for typhoidal Salmonellas in household and surface waters in Nepal identifies potential transmission pathways," has been formally accepted for publication in PLOS Neglected Tropical Diseases.

Best regards,

Shaden Kamhawi

co-Editor-in-Chief

Paul Brindley

co-Editor-in-Chief
